# Effectiveness of Bioinks and the Clinical Value of 3D Bioprinted Glioblastoma Models: A Systematic Review

**DOI:** 10.3390/cancers14092149

**Published:** 2022-04-26

**Authors:** Shye Wei Leong, Shing Cheng Tan, Mohd Noor Norhayati, Mastura Monif, Si-Yuen Lee

**Affiliations:** 1Department of Internal Medicine, School of Medical Sciences, Health Campus, Universiti Sains Malaysia, Kota Bharu 16150, Malaysia; shyeweileong@student.usm.my; 2UKM Medical Molecular Biology Institute, Universiti Kebangsaan Malaysia, Kuala Lumpur 56000, Malaysia; sctan@ukm.edu.my; 3Department of Family Medicine, School of Medical Sciences, Health Campus, Universiti Sains Malaysia, Kota Bharu 16150, Malaysia; hayatikk@usm.my; 4Department of Neuroscience, Central Clinical School, Monash University, Melbourne, VIC 3004, Australia; mastura.monif@monash.edu

**Keywords:** 3D bioprinting, bioinks, glioblastoma models, tumour microenvironment, cell cultures, drug screening, drug response

## Abstract

**Simple Summary:**

Glioblastoma is the most malignant cancer of the glioma series, and it is highly invasive. The progression and recurrence of glioblastoma remain common due to the development of drug resistance. Of the current disease models and strategies used in pre-clinical studies for drug testing, three-dimensional (3D) bioprinting is an emerging technology in constructing a glioblastoma model. In this paper, 19 out of 304 articles yielded from the database search were selected and analysed through a systematic process. The selected studies present the effectiveness of different bioinks, which were used to mimic the tumour microenvironment of glioblastoma in bioprinting. The clinical value of the 3D bioprinted glioblastoma models on the efficacy of treatments or drug response was evaluated.

**Abstract:**

Many medical applications have arisen from the technological advancement of three-dimensional (3D) bioprinting, including the printing of cancer models for better therapeutic practice whilst imitating the human system more accurately than animal and conventional in vitro systems. The objective of this systematic review is to comprehensively summarise information from existing studies on the effectiveness of bioinks in mimicking the tumour microenvironment of glioblastoma and their clinical value. Based on predetermined eligibility criteria, relevant studies were identified from PubMed, Medline Ovid, Web of Science, Scopus, and ScienceDirect databases. Nineteen articles fulfilled the inclusion criteria and were included in this study. Alginate hydrogels were the most widely used bioinks in bioprinting. The majority of research found that alginate bioinks had excellent biocompatibility and maintained high cell viability. Advanced structural design, as well as the use of multicomponent bioinks, recapitulated the native in vivo morphology more closely and resulted in bioprinted glioblastoma models with higher drug resistance. In addition, 3D cell cultures were superior to monolayer or two-dimensional (2D) cell cultures for the simulation of an optimal tumour microenvironment. To more precisely mimic the heterogenous niche of tumours, future research should focus on bioprinting multicellular and multicomponent tumour models that are suitable for drug screening.

## 1. Introduction

Glioblastoma (World Health Organisation (WHO) grade IV glioma) is the most common primary brain cancer. It is highly invasive and the most malignant of the glioma series of cancers [1,2]. Glioblastoma patients have a poor prognosis, with a median survival time of around 15 months for a newly diagnosed glioblastoma and 5–7 months for recurrent glioblastoma. The 5-year survival rate is as low as 6.8% [3,4] despite conventional treatment modalities. Surgery is the first line of treatment for glioblastoma. Maximal surgical resection of this cancer has been associated with longer progression-free survival (PFS) and overall survival (OS) [5]. Due to the highly invasive nature of this cancer, the presence of microscopic disease and limitations in the extent of debulking tumour resection are not curative. Following surgery, the patients are usually also given radiotherapy and chemotherapy.

Radiotherapy using a standard radiation dose of 60 Gy is commonly administered as a primary or follow-up treatment to further improve the PFS and OS. In addition, in newly diagnosed glioblastoma patients, concomitant use of temozolomide (TMZ), an oral alkylating drug, significantly improves the OS from 12.1 months (with radiation alone) to 14.6 months (radiotherapy plus TMZ) [6]. Despite this, and due to the development of TMZ resistance and TMZ’s inability to act on tumour stem cells, disease progression and recurrence remains common. Once glioblastoma recurrence occurs, there are limited therapeutic options available [7]. Nowadays, several advanced treatment strategies, such as molecular targeted therapy, immunotherapy, gene therapy, stem cell-based therapies and nanotechnology, are being contemplated as possible alternatives to the current glioblastoma treatment procedures [8]. However, these strategies are in the early phases of investigation, and their usefulness in treating glioblastoma patients remains to be elucidated.

In oncology clinical trials, drug development has a low success rate of only 3.4% [9]. This could be attributed to the widespread use of 2D monolayer cultures of cancer cells, which are overly simplistic versions of the in vivo microenvironment and are insufficient to mimic accurate tumour biology, immunology and physiology. Therefore, better models are needed to study glioblastoma and its responses to drugs. In recent decades, 3D bioprinting has been exploited for the development of improved cellular models for cancer research. The technology uses computer-assisted design (CAD) software to deposit bio-based material(s) or bioinks in a layer-by-layer fashion, with the aim of replicating the natural extracellular matrix (ECM) of human organs utilising biocompatible materials embedded with living cells and growth factors or hormones [10]. Notably, 3D bioprinting has the potential to address a variety of medical research problems and has been used in many different areas, including drug delivery, regenerative medicine and functional organ replacement.

In principle, an ideal bioink should have similar rheological/mechanical and biological properties as the targeted tissues (or organisms) to ensure the printed constructs mimic the original tissue/organ. Bioinks are a key element to achieve model shape preservation via easy mastering of the underlying crosslinked network structure, biocompatibility, biodegradability and chemical modification accessibility [11]. Furthermore, intercellular communication and efficient transport of bioactive chemicals or therapeutic substances can be aided by 3D bioprinting. As a result, by formulating bioinks specific for neuronal and cancerous tissues and spatially patterning relevant cell types and structural properties, the microenvironments can be controlled more precisely with 3D bioprinting methods compared to traditional 2D cultures, with the aim of reproducing in vivo glioblastoma characteristics in vitro [12].

Although 3D bioprinting has drawn a lot of attention in biomedical science in recent years, review papers in this field have focused mainly on specific aspects of the technology, such as its fundamental principles and techniques. However, more insights into the nature of the process, as well as the materials and technology used, biocompatibility and clinical application, particularly in cancer disease modelling, are important. The main objective of this systematic review is to determine the effectiveness of bioinks in mimicking the tumour microenvironment of glioblastoma and their clinical value when used as a disease model.

## 2. Materials and Methods

### 2.1. Search Strategy

The review was conducted in accordance with the Preferred Reporting Items for Systematic Reviews and Meta-Analyses (PRISMA) guidelines [13]. A comprehensive literature search was performed on five electronic databases, namely PubMed, Medline Ovid, Web of Science, Scopus and ScienceDirect. The keywords were (“3D bioprinting” OR “3D printed” OR “three-dimensional in vitro model”) AND (“glioblastoma” OR “glioblastoma multiforme”). This systematic review was registered in PROSPERO (313977).

### 2.2. Study Selection and Eligibility Criteria

The titles and abstracts of all the identified records were screened independently by two authors (S.W.L and S.C.T) to select potentially relevant studies based on pre-designated inclusion and exclusion criteria. Any disagreement was resolved by a third author (S.Y.L). The inclusion criteria were as follows: (1) used bioinks, (2) in vitro and in vivo studies, (3) type of glioblastoma cell lines or cells derived from glioblastoma and the investigations performed, (4) used 3D bioprinted scaffolds, and (5) original article written in the English language only. The exclusion criteria were as follows: (1) non-particular interest in 3D bioprinting, (2) systematic and narrative reviews, interpretations, case series, guidelines and technical reports.

### 2.3. Data Extraction

The full texts of the articles that met eligibility criteria were further reviewed, and data extraction was independently performed by two authors (S.W.L and S.C.T). The following data were extracted from the included studies: (1) study information (authors, year of publication and study design); (2) intervention details (biomaterials and cells used, crosslinking methods and materials, 3D bioprinting techniques and drug testing); (3) outcome details (rheological and morphological characteristics of bioinks, biological characteristics of glioblastoma cells such as cell viability or cell proliferation or cell migration, and drug response). Disagreements were resolved by discussion amongst the authors (S.W.L and S.C.T), with the advice of a third author (S.Y.L) when necessary.

### 2.4. Quality Assessment

The quality of the included studies was appraised independently by the same two authors (S.W.L and S.C.T) using the proposed checklist by the Joanna Briggs Institute (JBI) [14]. JBI is an international research organisation established at the University of Adelaide’s Faculty of Health and Medical Sciences in Australia to support evidence-based healthcare and research. The JBI critical appraisal tool was used to assess a study’s methodological quality and identify the possibility of bias in its design, conduct, and analysis [14]. Each item on the risk of bias tools was scored with A (indicating low risk of bias), B (indicating high risk of bias), C (indicating bias not clear) or D (indicating not applicable).

## 3. Results

Initially, the database search yielded 304 articles, but after removing the duplicates, only 264 remained. After reviewing the titles and abstracts, 159 records were excluded due to being irrelevant. The full texts of the remaining 105 publications were then retrieved and screened. Another 86 publications were removed due to not satisfying the eligibility criteria, leaving 19 studies for inclusion in this systematic review [3,8,15,16,17,18,19,20,21,22,23,24,25,26,27,28,29,30,31]. Figure 1 shows a flow chart of the search results with reasons for article exclusion.

### 3.1. Quality Evaluation

The risk of bias in the included studies was assessed using a checklist of the JBI [14]. In general, almost all studies had a low risk of bias. There were, however, two studies [23,26] that did not explicitly state whether there was a control group. One [27] study was suspected high risk of bias for the multiple outcome assessments taken before and after the intervention/exposure. Three studies [8,17,20] were not clear with reliable outcomes measured or without number of replicate experiment, and one study [20] was found unclear without method of statistical analysis. Follow-up was not applicable to all the included studies. The results of the risk assessment are summarised in Table 1.

### 3.2. Characteristics of the Included Studies

The characteristics of the included studies are shown in Table 2. Sixteen studies [3,8,17,18,19,20,21,22,23,24,25,26,27,28,29,30] conducted in vitro assays, and three studies conducted both in vitro and in vivo experiments [15,16,31].

#### 3.2.1. Cell and Animal Models

The human glioblastoma cell line U87 was used in the majority of in vitro research (47.4% of the included studies). Two studies used only human glioblastoma cell line U87 [8,26]; one study used human glioblastoma cell line U87 and glioma stem cell line SU3 [3]; one study co-cultured human glioblastoma cell line U87 with human astrocytes [27]; one study co-cultured glioblastoma cells U87 with human vascular endothelial cells (HUVECs) and lung fibroblasts (LFs) [17]; one study co-cultured human glioblastoma cell line U87 and glioblastoma stem cell lines (G7, G144, G166) with monocytic MM6 [30]; one study co-cultured normal U87 cells and GFP-expressing U87 with human cerebral microvascular endothelial cell line (hCMEC/D3) [20]; and one study involved human glioblastoma cell line U87vIII and neuroblastoma SK-N-BE(2) [21]. In two studies, the human glioma cell line U118 was co-cultured with the human glioma stem cell GSC23 [16,24]. In five studies, only one cell line was used, namely the human glioma cell line U118 [15], human glioma stem cell GSC23 [23], human glioblastoma cell U-251 [25], human glioblastoma cells D54-MG [18] and GL261 mouse glioblastoma cell line [19], respectively. In addition, a total of four studies utilised primary cells from the patients. For example, one study used patient-derived glioblastoma cells alone [29]; one study co-cultured human glioblastoma cell line U87 cells and patient-derived glioblastoma with HUVECs [22]; one study co-cultured human patient-derived glioblastoma stem cells (GSCs) TS576 with HUVECs [28]; and one study co-cultured patient-derived GSCs with macrophages, astrocytes and neural stem cells (NSCs) [31]. Additionally, there were three in vivo studies of 3D bioprinting which utilised 4–6 week old nude mice; however, the studies did not disclose the number of animals used [15,16,31].

#### 3.2.2. Bioinks, 3D Bioprinting and Crosslinking Methods

The bioinks used, along with their crosslinking methods, are presented in Table 2. Alginate made up the great majority of the bioinks in this systematic review (63.2% of the included studies). Four studies used sodium alginate or alginate solution alone [18,21,24,25]; one study used a combination of sodium alginate and gelatine [16]; two studies used a combination of fibrin, alginate and genipin [8,27]; one study used a combination of RGD-alginate (alginate conjugated with Arg-Gly-Asp peptide sequence), hyaluronic acid (HA) and collagen-1 [30]; two studies used a combination of gelatin, alginate and fibrinogen (GAF) [3,17]; and two studies used a combination of GAF and transglutaminase [15,23]. Aside from alginate, brain decellularised extracellular matrix (dECM) or collagen were used as bioink in one study [22]. Matrigel was utilised in one study [26]; poly (ethylene glycol) diacrylate (PEGDA) and Bisphenol A ethoxylate dimethacrylate (BPADMA) were used in another [29]; gelatin methacrylate (GelMA) was used in another study [19]; and GelMA and glycidyl methacrylate-HA (GMHA) were used in two other studies [28,31]. One study employed magnetically-responsive cage-like scaffolds (MRCSs) [20].

Extrusion is the most commonly used method in 3D bioprinting (52.6% of the included studies) [3,8,15,16,18,19,23,24,27,30], while several studies employed other printing methods, such as the methods of two-photon lithography [20], droplet-based bioprinting [21], glioblastoma-on-a-chip [22], melt electrowriting [26], projection micro-stereolithography [29], and digital light processing [28,31]. Various crosslinking methods were utilised, with the majority of research using calcium chloride as a crosslinking agent (57.9% of the included studies). Crosslinking methods included the following: chemical crosslinking with calcium chloride [16,21,24,25,30] or in combination with thrombin, transglutaminase or chitosan [3,8,15,17,23,27]; chemical crosslinking with calcium carbonate [18]; and photo crosslinking [19,20,28,29,31]. Nonetheless, two studies did not use any crosslinking agent [22,26].

### 3.3. Physical Properties and Biocompatibility Measures

The properties of bioinks and their impact on cell morphology, biological characteristics and drug response are shown in Table 3. The included studies reported the pore size of hydrogels, which varied from 2–400 µm, while the porous percentage ranged from 53–89%. The alginate bioink with pore sizes of 100–400 µm and a porosity of 89% was found to be able to preserve cell viability at around 78% even after being bioprinted for 21 days, whereas the GAF bioink with pore sizes of 338.41 ± 23.18 µm was found to be able to preserve cell viability at around 89% even after being cultured for 15 days [3,19,23,25,31].

Most of the natural-based bioinks were reported to have excellent biological properties. A total of 11 studies reported a minimum of 78% to more than 90% cell viability [3,8,15,16,17,21,22,23,24,25,30], and 13 studies reported on cellular events such as cell proliferation, migration and spheroid formation within the bioink scaffolds [3,8,19,20,21,22,23,24,26,27,28,30,31]. Three studies reported that the diameter of tumour spheroids ranged from 21.71 µm to around 250 µm [16,17,25], while patient-derived spheroids ranged from 100–300 mm and patient-derived organoids ranged from 400–600 mm [29]. For the studies involving animal models, one reported that tumours formed by 3D-cultured cells were larger than those formed by 2D cells after 42 days [15], and another reported that 3D-U118 and 3D-GSC23 tumours had an outer capsule on their surface with many blood vessels [16].

### 3.4. Drug Response

Among the 19 included studies, 9 studies used TMZ with concentrations ranging from 0–1600 µg/mL (47.4% of the included studies). Four studies focused on TMZ alone [3,15,24,28]; one study compared TMZ with cisplatin (CIS) [30]; and one study compared TMZ with abiraterone, vemurafenib, ifosfamide, erlotinib and gefitinib, respectively [31]. Three studies involved a combination of TMZ and other drugs (one study compared TMZ with sunitinib (SU) and the combination of TMZ + SU [17]; one study compared TMZ with BEZ235, niraparib, the combination of TMZ + BEZ235 and niraparib + BEZ235 [29]; and one study compared TMZ with CIS, and the combination of CIS + KU60019 (KU), CIS + KU + O^6^-benzylguanine (O^6^BG), TMZ + methoxyamine (MX), TMZ + O^6^BG and TMZ + O^6^BG + MX [22]). Other studies used doxorubicin alone [21]; cordycepin and doxorubicin [25]; carmustine (BCNU) combined with AS1517499 and BLZ945 [19]; antibody-functionalised nutlin-loaded nanostructured lipid carriers (Ab-Nut-NLCs) [20]; and compound 15 (N-cadherin antagonist) [27]. In summary, five studies found that 3D-cultivated cells had higher viability than 2D cultured cells and that the drug’s half-maximal inhibitory concentration (IC_50_) was higher in the 3D-cultured model than in the 2D-cultured model [3,15,19,25,30]. Two studies reported that cell viability decreased as the concentration of drug increased (e.g., TMZ and N-cadherin antagonist) [24,27]. Three studies found that combining drugs efficiently reduced cell viability [17,22,29]. Five studies reported that co-cultured or stiff models were more resistant to drugs than mono-cultured or soft models [19,24,27,28,31], and one of these studies further validated the efficacy of ifosfamide (80 mg/kg) using a subcutaneous xenograft mouse model [31]. The results showed that tumour growth was reduced, in line with the efficacy achieved when tested in the in vitro co-cultured model [31].

## 4. Discussion

### 4.1. Overview of the Included Studies

This systematic review of 19 studies showed that bioinks were effective in simulating the glioblastoma tumour microenvironment, and the models can contribute to a more accurate drug testing. Most 3D bioprinted cell culture models demonstrated excellent cell viability and cell proliferation within the scaffolds. Moreover, 3D bioprinted glioblastoma models showed higher resistance to drugs when compared with the conventional 2D cell cultures, indicating that the 3D bioprinted models represented the in vivo morphology and complex tumour microenvironment better than 2D cell cultures. In a 3D environment, co-culturing of a tumour with macrophages, astrocytes or HUVECs also resulted in higher cellular viability and drug resistance as compared to mono-culturing of a tumour.

3D printing is an emerging technology in brain cancer modelling and drug screening. In spite of increasing published literature that suggested the advantages of 3D bioprinting, most of the studies were pre-clinical, with small sample sizes. In addition, there is a lack of a comprehensive systematic review of the context of bioink materials, material characteristics and effects, and bioprinting strategy. To our knowledge, this is the first systematic review to assess the effectiveness of bioinks in the 3D bioprinting of glioblastoma models and their clinical value.

### 4.2. Bioink Materials and Combination

Alginate has been extensively utilised as a bioink with or without a combination with other matrix components [32]. Wang et al. reported that the combination of alginate and gelatin produced good shear-thinning properties, sufficient mechanical strength after crosslinking and excellent physicochemical properties [16]. The bioink scaffold had well-defined pores for nutrient and oxygen exchange, subsequently supporting the cell viability.

Cells have receptors but not for alginate. The alginate was chemically modified using carbodiimide conjugated to an Arg-Gly-Asp peptide sequence (RGD) to improve cell-cell interaction and matrix interaction [33,34]. When compared to unmodified alginate, RGD-alginate showed an expansion of cells and the formation of cell aggregate in less than 24 h. U87MG cells were able to retain high viability in RGD-alginate matrices and grew in culture for more than a month [30]. In addition, cell proliferation increased when the density of RGD increased [35,36].

Alginate has also been used in combination with fibrin and genipin [8,27], and the addition of fibrin in the hydrogels was found to enhance the expansion of stem cells and tumourigenic cells [37,38,39]. The polymerisation of a fibrin-based bioink with a combination of alginate and genipin was initiated by the enzyme thrombin. A chemical crosslinker, e.g., calcium chloride (CaCl_2_), is commonly used to crosslink alginate, which interacts with the calcium ion (Ca^2+^) binding sites of fibrin, while genipin interacts with the amine groups to enhance polymerisation and the degree of lateral aggregation and enables stabilisation of the fibrin network [40]. Genipin has also been demonstrated to crosslink fibrin [39] and chitosan [41], successfully slowing down scaffold degradation [8]. In addition, the combination of gelatin with alginate and fibrinogen (GAF) was introduced due to its contribution to high structural stability and suitability for long-term cell culture. Gelatin was selected because it featured cell-binding motifs and allowed physical crosslinking at low temperatures [17].

The ECM is made up of a variety of macromolecules that dictate the tissue’s unique biochemical and biomechanical properties, and it plays an important role in cancer progression. As it is difficult to optimally reproduce the intrinsic complexity of native ECM, decellularised ECM (dECM) is one of the options for producing a microenvironment similar to that of the parental tissue. Cancer cells seeded onto dECM have recently been shown to have elevated expression of genes associated with invasion and enhanced interactions between the cells and ECM molecules [42,43]. Porcine brain was successfully decellularised and formed a printable dECM bioink with a serial treatment of chemical and enzymatic agents. Glioblastoma cells showed a higher proliferation rate and enhanced invasion capability in the brain dECM gel compared to the glioblastoma cells in collagen gel. In addition, HUVECs in the brain dECM gel showed an increased expression of the genes related to cell junction molecules and ECM remodelling protein (matrix metalloproteinase 9). As such, tubule networks were formed more actively in the brain dECM gel than in the collagen gel over two weeks [22].

In healthy brain tissues, HA is the most abundant ECM component. HA promotes glioblastoma migration by regulating glioblastoma invasion via the receptor for hyaluronan-mediated motility (RHAMM) and CD44, as well as other mechanical and topographical signals [44]. Recent studies showed that a combination of 0.25% HA with gelatin methacrylate (GelMA) had successfully enhanced glioblastoma stem cells’ pluripotency and resistance [28,31]. The results were in line with the findings reported by Pedron et al. [45]. GelMA also served as a stiffness modulator, resulting in acceptable mechanical qualities with minimal biochemical cues [31].

### 4.3. Physical Properties of the Bioink Scaffolds

In bioprinting, the architectural design of a tumour model is critical. An interconnected porous structure of bioink is beneficial to mimic the vasculature for nutrient and gaseous exchange, allowing surrounding cells to maintain high viability and functionality. Chen et al. discovered that a 3D microenvironment generated by 3D porous scaffolds not only promoted the formation of tumour cell spheroids but also greatly increased the invasiveness and chemotherapeutic resistance of tumour cells cultivated on 3D scaffolds compared to 2D models [46]. Moreover, Druecke et al. found that the pore size of the scaffold is a key factor of scaffold vascularisation as they discovered that blood vessel ingrowth was significantly accelerated in those scaffold pores with a size larger than 250 µm compared to those with a size less than 250 µm [47]. Hence, optimal pore size is deemed crucial in the fabrication of a functional 3D cell culture system for effective glioblastoma disease modelling.

Shear-thinning is an important parameter in 3D bioprinting because a bioink with excellent shear-thinning quality can minimise clogging during the printing process and immediately restore the structural consistency after printing so that the next layer can be supported [48,49,50]. The shear force generated during printing can be reduced by using a suitable nozzle diameter and low-viscosity hydrogels [51]. For example, to reduce the shear force, cells were integrated with 10% gelatin bioink using a nozzle with a diameter of 0.26 mm at a controlled temperature of 25 °C, and the chamber temperature was lowered to 10 °C during printing to increase the shape fidelity of printed scaffolds [23].

Stiffness is another feature that indicates the ability of bioink scaffolds to withstand mechanical force. This parameter could be quantified by graphing the stress-strain curves of the scaffolds under mechanical pressure and computing the slope of the curve, also known as Young’s modulus [52]. The Young’s modulus is a critical parameter of biomaterials that influences cell proliferation and differentiation direction [53]. In a study performed by Chaicharoenaudomrung et al. the Young’s modulus of the alginate scaffold decreased over time [25]. It was found to be closer to that of the brain cancer tissue at around 7 kPa [54] than cell cultures on polystyrene plates (2–4 GPa) [55]. Tang et al. had reported that the stiffness of the GSC-encapsulated tumour core was 2.8 ± 0.6 kPa, whereas the less dense peripheral region was 0.9 ± 0.2 kPa (consisting of encapsulated neural progenitor cells and astrocytes) [31]. The stiffness of the peripheral region was meant to be similar to that of healthy brain tissue, which is claimed to be 1 kPa [44]. In another study conducted by Tang et al. two types of mechanical properties or stiffnesses were created to represent glioblastoma and healthy brain tissue in the ECM regions, respectively [28]. A bioprinted region with a stiffness of 21 kPa was referred to as the stiff model, while another bioprinted region with 2 kPa was referred to as the soft model, considering that the matrix stiffness in glioblastoma tissues could rise to 26 kPa from the previous clinical investigations [56,57,58,59]. The tumour cells and epithelial cells showed high viability in their respective hydrogel environment after one week of being cultured. Therefore, Tang et al. concluded that the stiffness-patterned models may be ideal for mimicking different stages of glioblastoma development because the tumour and endothelium regions were intended to have stiffness simulating their native states and both invasion patterns have previously been seen for glioblastoma cells [28].

### 4.4. Biocompatibility and Cellular Response

The biocompatibility of bioinks was thoroughly explored. When considering a possible material for medicinal usage, cytotoxicity should be considered. The live/dead cell assay was used in most of the studies to ensure cell-to-material chemical contact did not cause cytotoxicity. Multiple diverse populations of malignant and supportive stromal cells make up the brain tumours, and these intricate cellular interactions are critical for tumour survival, growth and progression. Intratumoural cell heterogeneity in glioblastoma is extensive, with contributions from astrocytes, neurons, macrophage/microglia and vascular components. The researchers also showed the advantage of using 3D bioprinted tumour models by bioprinting several different cell types simultaneously [17,22,27,28,30,31]. Bioprinted HUVECs and LFs in GAF hydrogel were cultured until blood vessels with lumens developed. The multicellular tumour spheroids were then seeded into the blood vessel layer and incubated until the blood vessel layer’s endothelial cells migrated into the multicellular tumour spheroids and displayed angiogenesis, while some cancer cells penetrated the blood vessel layer [17]. In another study, migration of HUVECs towards the glioblastoma cells was seen in 3D co-culture models. The migrating HUVECs in the stiff model had a sprouting blood vessel shape and were in close contact with the glioblastoma cells, whereas the HUVECs in the soft model had an expansive-growth morphology with no apparent sprouting. The expression of the angiogenic marker vascular endothelial growth factor (VEGF) in the tumour cells in the co-cultures of both the soft and the stiff hydrogel was significantly increased compared to the tumour-only models [28].

Furthermore, Heinrich et al. found that the tumour cells migrated towards macrophages as opposed to towards tumour cells themselves or to an empty well [19]. There was significant upregulation of glioblastoma markers in the co-cultured model with RAW264.7 macrophages. Therefore, the research demonstrated that tumour cells may recruit macrophages to their site and train them to maintain or enhance tumour survival and growth [19]. In another study conducted by Tang et al. the 3D tetra-culture model (macrophages were combined with GSCs within the tumour core surrounded by astrocytes and neural progenitor cells) showed an elevation of the glioblastoma tissue-specific gene set when compared to a GSC spherical culture [31]. The presence of macrophages enhances genes that would promote hypoxia and pro-invasive transcriptional profiles, demonstrating that the tetra-culture model or multicellular culture system recapitulates the transcriptional states seen in patient-derived glioblastoma tissues [31].

### 4.5. Drug Response and Clinical Value

The included studies found that 3D-cultivated models had higher cell viability and higher IC_50_ than 2D-cultured models. Wang et al. hypothesised that the combination of GSC proliferation, a hypoxic environment and the activation of epithelial-mesenchymal transition (EMT) resulted in increased drug resistance in 3D-cultured cells [15]. Cancer stem cells (CSCs) are not only responsible for chemoresistance [60] but also tumourigenicity. To evaluate the tumourigenicity of CSCs in nude mice, 1 × 10⁴ cells were harvested from 2D or 3D conditions, and 3D-cultured cells were shown to be more tumourigenic than 2D-cultured cells, indicating that the stemness qualities of glioma cells were improved in 3D bioprinted scaffolds [15].

Tang et al. reported that the stiff condition and co-culture with endothelial cells enhanced glioblastoma drug resistance as compared to a spherical culture control. Furthermore, the stiff co-culture model showed the highest tumour cell viability after TMZ treatment, whereas the soft model showed higher TMZ sensitivity, indicating that the cancer drug, TMZ, induced cell cycle arrest and halted cell division, thus inhibiting cell proliferation [28].

Shell-GSC23/core-U118 hydrogel microfibers showed higher resistance to TMZ with a significantly lower methylation rate of O^6^-methylguanine-DNA methyltransferase (MGMT) promoter when compared to core-U118 hydrogel microfibers [24]. Recent research has proven that MGMT is a key component in tumour prognosis [61,62]. Methylation of the MGMT promoter can silence the gene in cancer cells and limit their ability to repair DNA, rendering cancer cells more vulnerable to TMZ. The higher the degree of MGMT methylation, the lower the MGMT protein expression, resulting in a favourable prognosis in the setting of TMZ administration [24].

Various drugs and combinations have been used to evaluate treatment resistance and to identify a patient-specific drug combination [22]. The ataxia-telangiectasia mutant kinase, which activates critical proteins that initiate DNA-damage-response pathways, was inhibited by combining CIS with KU [63]. Surprisingly, CIS + KU reduced the survival cell percentage of glioblastoma-28-on-a-chips; however, glioblastoma-37-on-a-chips were less sensitive to the same treatment. When compared with glioblastoma-37-on-a-chip, glioblastoma-28-on-a-chip was more susceptible to O^6^BG (a pseudosubstrate of MGMT) and MX (a base excision repair pathway inhibitor), including the combination of O^6^BG + MX and the combination of CIS + KU + O^6^BG + radiation. The resulting ex vivo glioblastoma model can be used for the identification of an optimal treatment for patients with the aid of personal bioinformatics analysis [22].

Tumour cells in 2D and 3D models may respond differently to the same treatment. Furthermore, drug test findings acquired from animal models cannot fully reflect what would be observed in the human body due to cross-species variations, as more than 95% of drugs that are successful in animals are not as effective in humans [3]. In the end, the majority of drugs would fail in the pre-clinical testing. Hence, a 3D bioprinted tumour model, which gives a more accurate representation of the tumour microenvironment, is an ideal tool for evaluating drug efficacy.

### 4.6. Study Limitations

There are some limitations to this systematic review. First, no specific checklist has been developed for the analysis of the risk of bias in in vitro studies. Thus, we used the JBI checklist to assess the studies that were included. Moreover, studies published in languages other than English were excluded. Unpublished reports or studies published in languages other than English may be missed unintentionally. Furthermore, while the use of 3D bioprinting for cancer modelling has been widely researched in vitro, only a few studies have been conducted in vivo. This limits our understanding of the clinical value of 3D bioprinting in in vivo conditions. Furthermore, there was heterogeneity among the studies, as the duration of observation and measurements as well as the cell lines used varied across studies, which made it unsuitable to quantitatively combine the findings.

## 5. Conclusions

In this systematic review, both in vitro and in vivo studies demonstrated that bioinks have a strong potential to imitate the 3D microenvironment of the native brain tumour tissue while promoting cell viability and proliferation. The difference between the effects of therapeutic drugs on 2D and 3D cultures was reported. Monolayer or 2D cell culture models fail to capture the heterogeneity and complexity of the tumour microenvironment. However, 3D bioprinted models allow the incorporation of more than one cell type, simulate the 3D geometry of native tumour tissues, and supply accurate nutrition and oxygen gradients. Some studies reported an increase in the drug efficacy in 3D models with a printed porous structure or a vascular network because more regions or higher surface volumes of the tumour were subjected to the drug. In drug screening, a combination drug of TMZ, BCNU, doxorubicin (DXR) and cordycepin (COR) was reported to have higher IC_50_ when tested with the 3D bioprinted glioblastoma model than the 2D cell culture model. Furthermore, multi-drug treatments (e.g., TMZ + SU, TMZ + BEZ235, and niraparib (NIRA) + BEZ235) have shown a greater therapeutic response than single-drug treatments.

Overall, the recent developments on in vitro 3D bioprinted glioblastoma models presented in this paper have contributed to a better understanding of the fabrication techniques of bioinks, characteristics and effectiveness of bioinks, tissue bioprinting strategies and the discovery of potential treatment targets. In several studies, the combination of bioinks of different materials, including natural and synthetic materials, was explored to construct a desired structure that supported the tumour microenvironment. We suggest that future research should focus on bioprinting of co-cultured tumour models with a vascular component, astrocytes, and microglia/monocyte/macrophages to represent a more precise heterogenous tumour microenvironment. Since most recent studies still use commercial cell lines, primary cells and cancer stem cells derived from patients can also contribute to a more precise model and effective evaluation. Additionally, the physical parameters of bioinks, such as stiffness and porosity of the scaffolds, must be considered in order to model the true state of the ECM. A sophisticated 3D glioblastoma tumour model is essential for high-throughput drug screening to replace conventional animal trials and may drive the realisation of personalised treatment in the future.

## Figures and Tables

**Figure 1 cancers-14-02149-f001:**
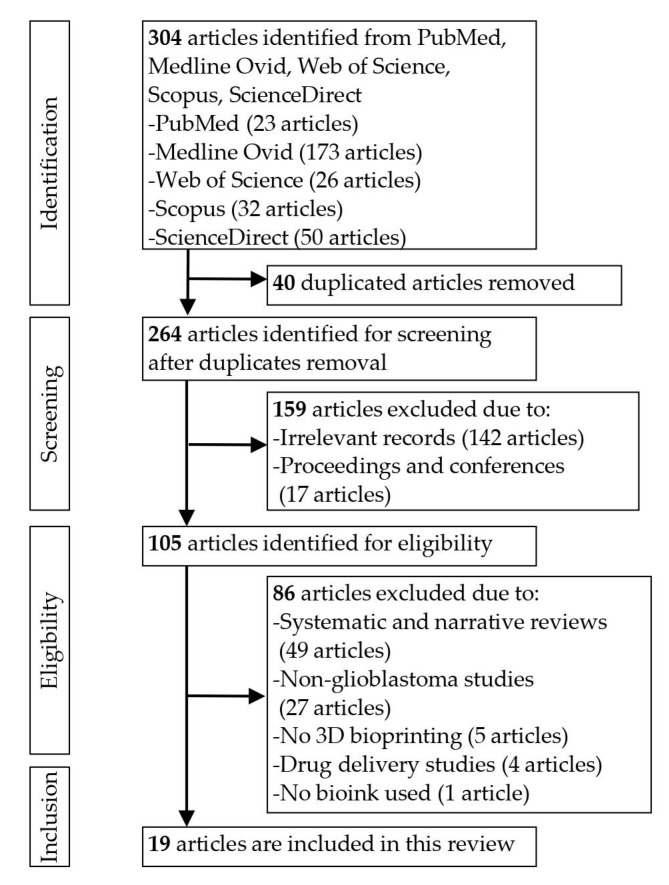
PRISMA flow chart.

**Table 1 cancers-14-02149-t001:** Risk of bias assessment of the included studies.

	References																		
**A: Low Risk of Bias** **B: High Risk of Bias** **C: Not Clear** **D: Not Applicable**	Wang et al. [15]	Wang et al. [16]	Dai et al. [3]	Han et al. [17]	Haring et al. [18]	Heinrich et al. [19]	Tricinci et al. [20]	Utama et al. [21]	Yi et al. [22]	Lee et al. [8]	Wang et al. [23]	Wang et al. [24]	Chaicharoenau-domrung et al. [25]	Ba-kirci et al. [26]	Smits et al. [27]	Tang et al. [28]	Chadwick et al. [29]	Hermida et al. [30]	Tang et al. [31]
**Checklist**																			
No confusion about which variable comes first	A	A	A	A	A	A	A	A	A	A	A	A	A	A	A	A	A	A	A
The subjects involved in any of the comparisons were comparable	A	A	A	A	A	A	A	A	A	A	A	A	A	A	A	A	A	A	A
Other than the exposure or intervention of interest, the subjects involved in any comparisons received similar treatment/care	A	A	A	A	A	A	A	A	A	A	A	A	A	A	A	A	A	A	A
There was a control group	A	A	A	A	A	A	A	A	A	A	C	A	A	C	A	A	A	A	A
Multiple outcome assessments taken before and after the intervention/exposure	A	A	A	A	A	A	A	A	A	A	A	A	A	A	B	A	A	A	A
Completed follow-up	D	D	D	D	D	D	D	D	D	D	D	D	D	D	D	D	D	D	D
Participants’ results measured in the same way in any comparisons	A	A	A	A	A	A	A	A	A	A	A	A	A	A	A	A	A	A	A
Reliable outcomes measured	A	A	A	C	A	A	C	A	A	C	A	A	A	A	A	A	A	A	A
Appropriate statistical analysis	A	A	A	A	A	A	C	A	A	A	A	A	A	A	A	A	A	A	A

**Table 2 cancers-14-02149-t002:** Characteristics of the included studies (*n* = 19).

Bioinks	Cells	Study Design	Printing Method	Crosslinking Methods	Drugs	Ref.
Gelatin, alginate, fibrinogen (GAF), transglutaminase	Human glioma cell line U118	In vitro and in vivo	Extrusion	Scaffolds were immersed in calcium chloride (CaCl_2_) solution for 3 min and then thrombin for 15 min after printing	TMZ	[15]
Sodium alginate and gelatin	Human glioma cell line U118 and human glioma stem cell GSC23	In vitro and in vivo	Extrusion	Scaffolds were immersed in CaCl_2_ solution for 3 min after printing	N/A	[16]
GAF	Glioma stem cell line SU3 and human glioblastoma cell line U87	In vitro	Extrusion	Transglutaminase was added to a hydrogel system. Scaffolds were first immersed in CaCl_2_ and then thrombin after printing	TMZ	[3]
GAF	Human glioblastoma cell line U87, human vascular endothelial cells (HUVECs) and lung fibroblasts (LFs)	In vitro	Not mentioned	Scaffolds were immersed in CaCl_2_ solution for 3 min and then thrombin for 15 min after printing	TMZ, sunitinib (SU)	[17]
Alginate solution	Human glioblastoma cells D54-MG	In vitro	Extrusion	Calcium carbonate was added to a hydrogel system	N/A	[18]
GelMA	Mouse glioblastoma cells GL261	In vitro	Extrusion	Photocrosslink	Carmustine (BCNU), AS1517499, BLZ945	[19]
Magnetically-responsive cage-like scaffolds (MRCSs)	Human glioblastoma cell line U87, GFP-expressing U87 and human cerebral microvascular endothelial cell line (hCMEC/D3)	In vitro	Two-photon lithography	Photocrosslink	Antibody-functionalised nutlin-loaded nanostructured lipid carriers (Ab-Nut-NLCs)	[20]
Sodium alginate	Neuroblastoma SK-N-BE (2) and human glioblastoma cell line U87vIII	In vitro	Droplet-based bioprinting	CaCl_2_	Doxorubicin	[21]
Brain decellularised ECM or collagen	Human glioblastoma cell line U87, patient-derived glioblastoma and HUVECs	In vitro	Glioblastoma-on-a-chip	No crosslinking applied	TMZ, cisplatin (CIS), KU60019 (KU), O^6^-benzylguanine (O^6^BG), methoxyamine (MX)	[22]
Fibrin, alginate, genipin	Human glioblastoma cell line U87	In vitro	Microfluidic extrusion	CaCl_2_, chitosan, thrombin were mixed and connected to the bioprinter through the ‘cross-linker’ pneumatic channel	N/A	[8]
GAF, transglutaminase	Human glioma stem cell GSC23	In vitro	Extrusion	Scaffolds were immersed in CaCl_2_ solution for 3 min and then thrombin for 15 min after printing	N/A	[23]
Sodium alginate	Human glioma cell line U118 and human glioma stem cells GSC23	In vitro	Extrusion	CaCl_2_ was used as printing receiving platform	TMZ	[24]
Sodium alginate	Human glioblastoma cells U-251	In vitro	Not mentioned	The scaffolds were crosslinked with 2% CaCl_2_ solution for 1 hr	Cordycepin, doxorubicin	[25]
Matrigel	Human glioblastoma cell line U87	In vitro	Melt electrowriting	No crosslinking applied	N/A	[26]
Fibrinogen, alginate, genipin	Human glioblastoma cell line U87 and human astrocytes	In vitro	Extrusion	CaCl_2_, chitosan, thrombin	Compound 15 (N-cadherin antagonist)	[27]
GMHA and GelMA	Human patient-derived GSCs (TS576) and HUVECs	In vitro	Digital light processing	Photocrosslink with rapid polymerisation of each region with 20–30 s of light exposure	TMZ	[28]
PEGDA and BPADMA	Patient-derived glioblastoma cells	In vitro	Projection micro-stereolithography	Photocrosslink by using phenylbis (2,4,6-trimethylbenzoyl) phosphine as a photoinitiator to initiate polymerisation and Sudan I as a photo absorber to control UV light penetration	TMZ plus BEZ235 or niraparib plus BEZ235	[29]
RGD-alginate, HA and collagen-1	Human glioblastoma cell line U87, monocytic (MM6), glioblastoma stem cell line (G7, G144 and G166)	In vitro	Extrusion	CaCl_2_ was used as a crosslinking agent for 3 min	CIS and TMZ	[30]
GelMA and GMHA	Patient-derived GSCs, macrophages, astrocytes, and neural stem cells (NSCs)	In vitro and in vivo	Digital light processing	Photocrosslink with exposure time of 20 s for the core and 15 s for the periphery	Abiraterone, vemurafenib, ifosfamide, erlotinib, gefitinib, TMZ	[31]

GAF: gelatin, alginate, fibrinogen; GBM: glioblastoma; GelMA: gelatin methacrylate; HA: hyaluronic acid; GMHA: glycidyl methacrylate-hyaluronic acid; dECM: decellularised extracellular matrix; MRCSs: magnetically-responsive cage-like scaffolds; PEGDA: poly (ethylene glycol) diacrylate; BPADMA: Bisphenol A ethoxylate dimethacrylate; RGD-alginate: alginate modified by Arg-Gly-Asp peptide sequence; HUVECs: human vascular endothelial cells; LFs: lung fibroblasts; hCMEC/D3: human cerebral microvascular endothelial cell line; GSCs: glioblastoma stem cells; NSCs: neural stem cells; CaCl_2_: calcium chloride; TMZ: temozolomide; SU: sunitinib; BCNU: carmustine; Ab-Nut-NLCs: antibody-functionalised nutlin-loaded nanostructured lipid carriers; CIS: cisplatin; KU: KU60019; O^6^BG: O^6^-benzylguanine; MX: methoxyamine.

**Table 3 cancers-14-02149-t003:** Bioink properties and the experimental outcomes of the studies included (*n* = 19).

Bioinks	Physical Characteristic	Cell Morphology	Biological Characteristic	Drug Response	Ref.
Gelatin, alginate, fibrinogen (GAF), transglutaminase	N/A	In vivo: Tumours formed by 3D cultured cells were larger than those formed by 2D cells on day 42	Cell viability: Day 0: 89.06 ± 3.58% Day 15: 84.30 ± 2.67%	After treatment with TMZ, the viability of cells in 2D and 3D culture began to decrease; 3D cultured cells showed higher viability	[15]
Sodium alginate and gelatin	N/A	Diameters of tumour cell spheroids formed: GSC23: 27.13 ± 2.59 μm U118: 21.71 ± 1.43 μm In vivo: 3D-U118 and 3D-GSC23 had an outer capsule on their surface with numerous blood vessels.	Cell viability: U118: 2h: 84.28 ± 2.15% Day 15: 85.36 ± 1.82% GSC23: 2h: 83.79 ± 3.08% Day 15: 87.85 ± 2.32%	N/A	[16]
GAF	Scaffold swelling ratio: Crosslinked by TG: 518.18 ± 60.58% Crosslinked by CaCl_2_: 501.85 ± 62.31% Pore diameter: 2–4 µm	Cells developed into spheroids after three weeks and pushed the surrounding hydrogels aside to take up more space within the scaffolds.	Live/dead cell ratio: 86.92%	Growth rates (1600 µg/mL TMZ for 48h): SU3: 3D: 107.20 ± 4.94% 2D: 72.73 ± 3.38% U87: 3D: 87.85 ± 4.57% 2D: 39.07 ± 3.57%	[3]
GAF	Storage moduli: 0.7–9 kPa Loss moduli: 0.06–1.7 kPa	Average diameter of multicellular tumour spheroids after 3 days: ~250 µm.	The viability of the HUVECs and LFs: >90% after bioprinting, >80% on day seven.	TMZ or SU significantly reduced the size of MCTS. Both TMZ and SU together further reduced the tumour size.	[17]
Alginate solution	Penetration time: 25–100 ng/mL EGF: 2.8–2.5 h 50–400 μM BK: 0.5–0.2 h	N/A	46% more glioblastoma cells migrate toward EGF. EGF: 20.7 ± 3.2% BK: 14.2 ± 2.9%	N/A	[18]
GelMA	G′: 1000 Pa; storage modulus: 10–20 Pa. G′ and G″ remained relatively stable with increasing shear rate. Average pore size: |17.08 ± 6.7 µm	Compared to empty wells, tumour cells showed a significantly higher migration toward RAW264.7 macrophages	Both RAW264.7 and GL261 cells remained viable for days 10 post-printing, and the cell-laden constructs displayed high metabolic activity.	IC_50_ of BCNU: 2D cell culture: 139 µM 3D mono-cultured GL261 cells: 581 µM 3D co-cultured RAW264.7: 887 µM. Tumours isolated from co-cultured treated with BLZ945, but not with AS1517499, showed slow growth.	[19]
Magnetically-responsive cage-like scaffolds (MRCSs)	N/A	One spheroid developed per MRCS after 5 days of growing	Immunofluorescence analysis against Ki-67 marker: the external layers of cells were in the interphase of the cell cycle, while the inner part cells were quiescent	About 70% of the GB cells inside the microcage were positive for ethidium homodimer-1.	[20]
Sodium alginate	N/A	Cells formed in a dense ball by using an individual alginate droplet	Cell viability: immediately pre-print and post-print: >98% after 72 h: >95% Percentage of cleaved caspase-3-positive cells: 3D bioprinted: 0.4% manual spheroid:1.7%	IC_50_ of doxorubicin: 1.06 to 1.48 mM	[21]
Brain decellularised ECM or collagen	N/A	GBM-28 cells showed increased invasion and a more spindle-like morphology in the BdECM gel than the collagen gel.	Both hydrogels demonstrated >90% cell viability, but proliferation was higher in the BdECM gel than the collagen gel after 10 days.	Survival percentage: GBM-28: CIS < TMZ GBM-37: slight decrease after CIS treatment Responsive to drug: CIS + KU, O^6^BG + MX, CIS + KU + O^6^BG with radiation: GBM-28-on-a-chip > GBM-37-on-a-chip O^6^BG was the most effective in suppressing the GBMs-on-chips.	[22]
Fibrin, alginate, genipin	N/A	Cells tended to form spheroids within the scaffolds and tended to grow in size and density with increased time within the scaffold.	Live/dead imaging: 88.78% ± 2.92% (post-printing) Cell viability: Day 1: 98.09% ± 0.89% Day 6: 91.78% ± 5.96% Day 9: 83.93% ± 5.75% Day 12: 86.12% ± 5.09%	N/A	[8]
GAF, transglutaminase	Pore size: 338.41 ± 23.18 μm Filament diameter: 324.27 ± 30.98 μm	Cells in 3D scaffolds gradually proliferated to form spheroids with full, uniform shapes and pushed the surrounding hydrogels away	Cell viability: after bioprinting: 86.27 ± 2.41% Day 15: 89.39 ± 1.86%	N/A	[23]
Sodium alginate	N/A	Core-U118 cells gradually proliferated into spheroids connected with each other until the formation of fiber-like cell aggregates.	Shell-GSC23/core-U118 (G/U) hydrogel Cell survival rate: 2 h: 93.72 ± 2.51% 15 days: 90.63 ± 1.54%	As the concentration of TMZ increased, the cell viability decreased gradually. G/U cultured cells showed greater viability than U microfiber-cultured cells.	[24]
Sodium alginate	Young’s modulus (kPa): Day 0: 131.0 ± 16.2 Day 7: 100.6 ± 9.6 Day 14: 73.2 ± 2.1 Day 21: 27.8 ± 7.4 Pore diameter: 100–400 μm Porosity: 89.5%	Spheroid diameter: Day 7: over 50% spheroids < 50 μm in diameter Day 14: up to 85% spheroids > 50 μm Day 21: 100% spheroids > 50 μm	Live/dead cells percentage: Day 7: 90.37 ± 1.76%, Day 14: 83.45 ± 3.79%, Day 21: 78.25 ± 5.11%	IC_50_ of doxorubicin: 2D: 1.98 ± 0.01 µg/mL 3D: 10.00 ± 1.0 µg/mL IC_50_ of cordycepin: 2D: 103.66 ± 10.26 µg/mL 3D: 207.33 ± 16.62 µg/mL	[25]
Matrigel	Elastic modulus: 4.5 mg/mL: 31 ± 5.6 Pa 6 mg/mL: 48 ± 9.2 Pa 8 mg/mL: 66 ± 4.4 Pa	Cell migration: 2 mg/mL: 1.9 ± 0.2 mm 4 mg/mL: 2.4 ± 0.5 mm 6 mg/mL: 3.2 ± 0.4 mm 8 mg/mL: 2.6 ± 0.4 mm	Cell percentage: 2 mg/mL: 14 ± 2.8% 6 mg/mL: 33 ± 6.3% 8 mg/mL: 31.2 ± 8.4%	N/A	[26]
Fibrinogen, alginate, genipin	N/A	Human glioblastoma cell U87 formed spheroids within the scaffolds after 6 days in culture.	N/A	3D-printed glioblastoma cell viability: 1mM: 86.5 ± 6.9% 5mM: 54.1 ± 8.9% 10mM: 50.6 ± 2.8% 25mM: 50.1 ± 3.6% 50mM: 46.7 ± 9.4% Cell viability of co-cultures: 5 mM: day 16: 82.6 ± 14.8% day 22: 68.5 ± 2.4% day 30: 16.8 ± 1.0% 10 mM: day 16: 83.1 ± 3.9% day 22: 25.0 ± 2.4% day 30: 11.1 ± 1.0%	[27]
GMHA and GelMA	Mean stiffness: Stiff model: 21 kPa Soft model: 2 kPa Pore size: stiff ECM < soft ECM	HUVEC morphology: Stiff model: sprouted blood vessels and close contact with glioblastoma cells. Soft model: expansive growth without visible sprouting.	Hypoxia-related genes and hypoxia-associated angiogenesis markers upregulated in the stiff condition. No significant difference in proliferation marker MKI67 of cells between both models. More KI67-positive cells were present in the soft model.	IC_50_ of TMZ: sphere cultured TS576 cells: 30 × 10−^6^ M TMZ treatment: No significant difference in the cell viability between both models. TMZ treatment in co-culture condition: Cell viability significantly increased in stiff models but not in the soft model	[28]
PEGDA and BPADMA	Good in shape programming and recovery	PDSs: 100–300 mm PDOs: 400–600 mm	NESTIN-expressing cells in the outer rim coexpressed other GSC markers widely detected in PDSs	Combination therapy increased apoptosis in GBM#46 PDOs and could significantly reduce migration and invasion of GBM-PDO cells	[29]
RGD-alginate, HA and collagen-1	Mean stiffness (kPa): 10 mM CaCl_2_: 11.9 50 mM CaCl_2_: 25.7	Cell spreading and apparent adhesion in <24 h within RGD-alginate	Cell viability: glioblastoma cells: >90%G144, G166, G7: >90%	IC_50_ of TMZ: U87: 1994 ± 1.0 μM| G7: 748.8 ± 1.1 μM (2-fold higher than 2D cell culture) IC_50_ of cisplatin: U87: 69.8 ± 1.1 μM G7: 241 ± 1.1 μM	[30]
GelMA and GMHA	Stiffness: tumour cell core: 2.8 ± 0.6 kPa NPCs and astrocytes peripheral: 0.9 ± 0.2 kPa Porosity: 53% Pore size: 85 μm	N/A	Cells showed increased levels of the proliferative marker Ki67 and increased protein expression of the stemness markers OLIG2 and SOX2.	GSC23 showed increased resistance to erlotinib, gefitinib, and TMZ in any 3D model than in sphere cultures. Tetra-culture GSCs showed increased sensitivity to abiraterone and ifosfamide than GSCs triculture, while vemurafenib sensitivity remained unchanged. In vivo: Ifosfamide therapy reduced tumour growth	[31]

GAF: gelatin, alginate, fibrinogen; GBM: glioblastoma; GelMA: gelatin methacrylate; HA: hyaluronic acid; GMHA: glycidyl methacrylate-hyaluronic acid; dECM: decellularised extracellular matrix; MRCSs: magnetically responsive cage-like scaffolds; PEGDA: poly (ethylene glycol) diacrylate; BPADMA: Bisphenol A ethoxylate dimethacrylate; RGD-alginate: alginate modified by Arg-Gly-Asp peptide sequence; G/U: Shell-GSC23/core-U118; EGF: epidermal growth factor; BK: bradykinin; HUVECs: human vascular endothelial cells; LFs: lung fibroblasts; NPCs: neural progenitor cells; PDSs: patient-derived spheroids; PDOs: patient-derived organoids; CaCl_2_: calcium chloride; TMZ: temozolomide; SU: sunitinib; BCNU: carmustine; Ab-Nut-NLCs: antibody-functionalised nutlin-loaded nanostructured lipid carriers; KU: KU60019; O^6^BG: O^6^-benzylguanine; MX: methoxyamine; CIS: cisplatin.

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
