# Peer review of "Effectiveness of Bioinks and the Clinical Value of 3D Bioprinted Glioblastoma Models: A Systematic Review"

_cancers, 2022, doi:10.3390/cancers14092149_

Round 1
Reviewer 1 Report
The authors present a very intriguing systematic review on bioprinted GBM models and the translational potential impact. This very well-written review presents several points of strength, and I think that it could be really interesting for suggesting new paradigms in neuro-oncology
I just only suggest moving Table 3 (Risk of bias assessment of the included studies) as supplementary material, due to its findings, just to improve the readers' experience
Author Response
Please see the attachment and revised manuscript.

Reviewer 2 Report
General comments:
The Systematic Review entitled "Effectiveness of Bioinks and the Clinical Value of 3D Bioprinted Glioblastoma Models: A Systematic Review" has exhaustively reviewed all the research articles regarding the field of interest. Authors have well structured the manuscript in Introduction, Materials & Methods, Results and Discussion; as expected for these kind of manuscripts, they have correctly represented a PRISMA flow chart following the criteria of inclusion for articles in the research field. Also, two table compiling exhaustive information about each of the 19 included studies, as well as interesting information about their developed bioinks and biocompatibility outcomes, are exposed.
This study provides relevant insights regarding the application of 3D bioprinting for developing glioblastoma models and their assessment as clinical value, which could be of great interest for the scientific community in the field of cancer. However, some concerns need to be addressed.
The major concerns are:
- From my point of view, it makes more sense to present the Quality Evaluation of the selected papers (current 3.4 section in line 248) at the beginning of the Results section, more than at the end of the results section. In this way, authors can explain the risk of bias of each of the selected articles that fulfilled the PRISMA flow chart.
- In my opinion, the results section could be improved. Currently, authors only describe the number of articles presenting a specific characteristic. I miss some percentages overall in 3.1.1 and 3.1.2 subsections. For example, in 3.1.2 subsection, line 176, "Alginate made up the great majority of the bioinks", or in line 188 "Extrusion is the most commonly used method", or also "the majority of research using calcium chloride as a crosslinking agent" in line 193. Showing percentages (x% of the studies) would facilitate the overall comprehension of the presented data. The same occurs when analyzing "Drug Response" data in section 3.3.
- Authors should explain in results section what does the in vivo study consist in line 245. As it is, it does not show what authors want to transmit to the reader. Why talking about in vivo if the manuscript is related to 3D bioprinting? explain the study.
- Discussion section should follow a comprehensive flow. I would recommend to guide the reader. Currently, authors make a good general vision/description in the two first paragraphs, but then jump from alginate to extracellular matrix, to the mechanical properties of the scaffolds, then biocompatibility, cell viability and drug resistance/efficacy, without following a logical thread. This makes discussion section hard to understand and to draw relevant ideas.
The minor comments are:
- In the Abstract, I think that Leong et al. should change 304 to 264 in line 16, since the existing studies are in fact 264 and 40 were duplicates. If preferred, authors could state "19 out of the 304 articles yielded from the database search".
- Authors should revise the manuscript to describe all the abbreviations (i.e WHO) and apply the abbreviations use throughout the manuscript, overall, the word glioblastoma (GBM). In this same line, cell lines naming is a bit confuse, especially U87MG in Table 1. Sometimes it is called "human glioma cell line U87" [3], other" GBM cells U87MG" [17], "U-87 MG cells" [22],.... I suggest to standardize cell lines names in Table 1 and all throughout the manuscript.
- Leong et al. should revise the references format in the text throughout the manuscript. For example, in lines 136-137, they should appear as [3,8, 15-31], in line 142, as [3,8,17-30]. In line 207 the order of references is not correct. Please revise all and correct.
- Title of subsection 2.1 should be corrected to "Search Strategy" instead of "Stratery" in line 95
- The sentence from line 188 to line 192 is not well explained. I guess authors would mean "while several studies employed other printing methods such as the method of two-photon litography, drop-on-demand,...
- Reference [21] method of bioprinting is droplet based bioprinting. Please correct in line 190.
- As subsection 3.1.2 is about bioinks composition, methods of bioprinting and crosslinking, maybe it could be entitled as "Bioinks, 3D printing and Crosslinking Methods"
